# Identification of Let-7 miRNA Activity as a Prognostic Biomarker of SHH Medulloblastoma

**DOI:** 10.3390/cancers14010139

**Published:** 2021-12-28

**Authors:** Maximillian S. Westphal, Eunjee Lee, Eric E. Schadt, Giselle S. Sholler, Jun Zhu

**Affiliations:** 1Sema4, 333 Ludlow St., Stamford, CT 06902, USA; maximillian.westphal@sema4.com (M.S.W.); eunjee.lee@mssm.edu (E.L.); eric.schadt@sema4.com (E.E.S.); 2Department of Genetics and Genomic Sciences, Icahn School of Medicine at Mount Sinai, 1 Gustave L. Levy Pl, New York, NY 10029, USA; 3Helen DeVos Children’s Hospital, Grand Rapids, MI 49503, USA; Giselle.SaulnierSholler@helendevoschildrens.org; 4College of Human Medicine, Michigan State University, Grand Rapids, MI 49503, USA; 5The Tisch Cancer Institute, Icahn School of Medicine at Mount Sinai, 1 Gustave L. Levy Pl, New York, NY 10029, USA

**Keywords:** miRNA, medulloblastoma, prognostic, MYCN, let-7, sonic hedgehog, ActMiR

## Abstract

**Simple Summary:**

Medulloblastoma is the most common malignant pediatric brain tumor. It can be divided into four molecular subgroups with clear biological and clinical differences: Group 3, Group 4, SHH, and WNT. The Group 3 subgroup has the lowest overall survival rate, and the WNT subgroup has the highest. It is known that MYCN and let-7 play a critical role in medulloblastoma tumorigenesis and progression. By integrating multi-omics data, including gene expression, methylation, copy number variation, and miRNA expression, we further divided the SHH subgroup according to MYCN expression and let-7 activity and found that the combination of high MYCN expression and high let-7 activity is associated with worse overall survival.

**Abstract:**

Medulloblastoma (MB) is the most common pediatric embryonal brain tumor. The current consensus classifies MB into four molecular subgroups: sonic hedgehog-activated (SHH), wingless-activated (WNT), Group 3, and Group 4. MYCN and let-7 play a critical role in MB. Thus, we inferred the activity of miRNAs in MB by using the ActMiR procedure. SHH-MB has higher MYCN expression than the other subgroups. We showed that high MYCN expression with high let-7 activity is significantly associated with worse overall survival, and this association was validated in an independent MB dataset. Altogether, our results suggest that let-7 activity and MYCN can further categorize heterogeneous SHH tumors into more and less-favorable prognostic subtypes, which provide critical information for personalizing treatment options for SHH-MB. Comparing the expression differences between the two SHH-MB prognostic subtypes with compound perturbation profiles, we identified FGFR inhibitors as one potential treatment option for SHH-MB patients with the less-favorable prognostic subtype.

## 1. Introduction

Medulloblastoma (MB) is the most common malignant embryonal brain tumor in children [1,2]. The 5-year overall survival rate for patients with medulloblastoma is 65–70% [3]. Treatment for medulloblastoma usually consists of a combination of surgery, radiation, and chemotherapy [4]. The risk stratification for patients is currently based first on age, because patients under 3 years old do not undergo craniospinal radiation therapy, and then on a combination of tumor size, histology, and metastatic disease [5,6,7]. The World Health Organization classifies medulloblastoma, which is very heterogeneous, into four molecular subgroups with clear biological and clinical differences: sonic hedgehog-activated (SHH), wingless-activated (WNT), Group 3, and Group 4 [8,9,10,11,12,13,14]. Group 3 has the worst overall survival rate, and WNT has the best overall survival rate [3,10]. These subgroups reflect different cells of origin for the tumor cells [9,15]. Furthermore, integrative genomic analyses revealed distinct molecular features of the four medulloblastoma subgroups, such that MB samples can be clustered into subgroups using features such as transcription, methylation, and copy number variation (CNV) [10,16,17]. For example, in addition to each subgroup having unique expression and methylation profiles, WNT and SHH MB have distinct expression and methylation profiles from Group 3 and Group 4 [8,16,17].

MB is generally treated with surgery followed by chemotherapy and/or radiation therapy. Given the aforementioned molecular differences among the subgroups, subgroup-specific targeted therapies have been investigated, such as SMO inhibitors in SHH-MB, which is known to have alterations to SUFU, GLI2, and MYCN [18,19]. In addition to differences among the subgroups, there is also a clear heterogeneity within the subgroups themselves [16,20], and SHH is the most heterogenous subgroup in terms of genomic, molecular, and clinical features [16,21]. Furthermore, SHH-MB tumors with MYCN amplification are generally resistant to SMO inhibitors [18]. The TP53 mutation status is a prognostic factor in MB and SHH-MB in particular [12,22,23,24,25]. TP53 mutations occur more frequently in the WNT and SHH subgroups (13.8% and 7.6%, respectively) vs. Group 3 and Group 4 (1.5% and 0%, respectively) [12]. It is hypothesized that TP53 mutations confer radiation resistance, so additional therapies are urgently needed to target high-risk patients in each individual MB subgroup [12,22,23,24,25].

The MYCN/LIN28/let-7 axis, downstream of SHH, plays a critical role in SHH-MB [16,17,26,27,28]. A high LIN28B expression has previously been shown to lead to a poor prognosis in neuroblastoma, Group 3, and Group 4 MB [29,30]. MYCN is normally highly expressed in undifferentiated cells and embryonal tumors, which are thought to be initiated by proliferating progenitor cells unable to differentiate [31,32,33]. For example, in newborn mice, high levels of Mycn were found in the brain, and these expression levels decreased into adulthood [31,34], and human infants are more likely to have SHH-MB over other MB subgroups [16]. In addition, the let-7 miRNA family, a group of well-known tumor-suppressing miRNAs [35,36], is a repressor of MYCN and can be repressed by LIN28B [37,38]. MicroRNAs (miRNAs) post-transcriptionally regulate genes and play a critical role in carcinogenesis and tumor progression [35,39,40,41]. However, the miRNA expression level does not directly reflect miRNA activity [42,43,44,45], so we developed ActMiR, a computational tool for inferring the activity of miRNAs in vivo based on changes in expression levels of target genes [46].

In this study, we investigated let-7 miRNA activity and MYCN interactions in MB. Since integrative analysis between methylation, transcription, and CNV can cluster MB into subgroups and because MB tumors consist of both stromal and tumor cells [47], we investigated whether the tumor cellular composition can be estimated based on subgroup-specific genomic features. We further investigated whether these cellular composition estimates, or together with let-7 miRNA activity, are related to MB prognosis and the drug response in MB subgroups.

Notably, we identified let-7 miRNA activity as a potential prognostic biomarker for SHH-MB. We further demonstrated that MYCN expression, in combination with let-7 activity, stratified patients in SHH-MB into subtypes associated with different overall survival. SHH-MB patients with both high let-7 activity and high MYCN expression had significantly worse survival than other SHH-MB patients. Our results suggest that the MB patients in each molecular subgroup are still heterogeneous and that a miRNA-mediated regulatory network can be used for dissecting heterogeneity and identifying novel subtype-specific prognostic markers and therapeutic targets.

## 2. Materials and Methods

### 2.1. Preprocessing of Gene Expression, miRNA Expression, Methylation Data, and CNV Data

We downloaded 763 MB mRNA expression profiles (GSE85217) and 763 matching MB methylation profiles (GSE85212) from the super series GSE85218 [16,48]. We also downloaded 285 MB mRNA expression profiles (GSE37382) and 1097 MB genotype profiles (GSE37384) from the super series GSE37385, which had some overlap with GSE85218 the dataset that was not listed in the meta data [17]. For inferring the activity of miRNAs with ActMiR, we downloaded the training dataset consisting of 73 mRNA expression profiles and 64 miRNA pediatric brain cancer expression profiles (GSE42658), including MB; of these data, 57 mRNA and miRNA profiles originated from the same non-control samples [49]. From this dataset, we used 14 pilocytic astrocytoma samples, 14 ependymoma samples, 9 MB samples, 5 glioblastoma samples, 5 atypical teratoid/rhabdoid tumor samples, 4 choroid plexus papilloma samples, 3 diffuse astrocytoma samples, 2 anaplastic astrocytoma samples, and 1 papillary glioneuronal sample. The 9 MB samples consisted of 5 Group 4 samples, 2 SHH samples, and 2 WNT samples.

All mRNA and miRNA expression data were log2 transformed. Copy number variation (CNV) was called on for the GSE37384 genotype profiles using version 1.64.0 of the R package DNACopy’s circular binary segmentation method [50]. For the GSE85212 methylation profiles, we extracted DNA methylation values (β values) for each probe. In the case of multiple probes mapping to the same gene, we performed a Spearman’s correlation with the gene expression and the probe, with the best *p*-value selected because methylation near the promoter regions is associated with gene repression [51,52,53]. To match the genotype data from GSE37384 to methylation and gene expression data from GSE85218, we used the tool MODMatcher to determine which samples were the most correlated and then used clinical data such as age and sex to confirm their identities [53]. This resulted in 229 samples with gene expression, methylation data, and CNV data. The results of this mapping and the original dataset sample names are presented in Appendix A.

### 2.2. Identifying Cis-Regulatory Genes by Integrating Gene Expression, Methylation Data, and CNV Data

To determine the cis-regulatory genes of MB, we used 229 samples with gene expression, methylation data, and CNV data by mapping between GSE85218 and GSE37384. We performed multiple linear regression on these 229 mapped samples as follows: $Exp_{g}\sim Methyl_{g}+CNV_{g}$, where $Exp_{g}$ indicates the expression levels of genes g, $Methyl_{g}$ indicates the DNA methylation level in a gene’s g promoter region, and $CNV_{g}$ indicates CNVs that contained a gene g in cis form. The DNA methylation level was rank-based inverse normal transformed. Cis-regulatory genes were determined based on the false discovery rate (FDR) 5% corresponding to *p*-values < 1 × 10^−7^. To calculate the FDR rates based on *p*-values, the multipletests function in the statsmodels package in Python with the Benjamini and Hochberg method was used [54]. We defined cis-Methyl genes as genes with a significantly negative coefficient for $Methyl_{g}$ variable in a multiple linear regression, cis-CNV genes as those with a significantly positive coefficient for the CNV variable, and cis-CNV/Methyl as the unique subset that met both criteria. At FDR 5% corresponding to *p*-values < 1 × 10^−7^, we identified 3630 cis-CNV, 589 cis-methyl, and 107 both cis-CNV and cis-methyl genes. We used a principal component analysis (PCA) to see how well these cis-regulatory genes can separate MB subgroups.

### 2.3. Identifying Subgroups in GSE42658

Since GSE42658 did not list the subgroup annotation of the MB samples, we performed subgroup classification as follows. Using the expression in GSE85218 of the cis-Methyl genes identified above (Methods 2.2), we iteratively took the mean difference of each of the four subgroups (SHH, WNT, Group 3, and Group 4) against a combined set of the other three. Next, we performed a Spearman’s correlation of each GSE42658 sample’s cis-Methyl expression against each of the subgroup’s one vs. all mean difference. The subgroup with the highest correlation was annotated as the GSE42658 sample’s subgroup.

### 2.4. Inferring Tumor Purity of SHH-MB Tumors

The expression of the cis-Methyl genes reflects a different cell of origin for each MB subgroup. To further refine SHH-MB subgroup-specific up/downregulated genes, we compared single-cell RNA-seq expression profiles from 25 MB patients (GSE119926) [55], including 23 diagnostic samples and two recurrences. We downloaded TPM values (transcript per million reads) for each gene and performed log2 transformation. Among 589 cis-Methyl genes, 247 and 248 were SHH-MB-specific up/downregulated, with genes expressed higher/lower in SHH-MB than the other subgroups at *p* < 0.001, respectively. Within the 247 SHH-MB-specific upregulated genes, 13 genes were expressed specifically in SHH-MB tumor cells (*p* < 0.001 and expression in non-SHH-MB tumor cells < 1.5), referred to as SHH-MB tumor cell-specific expressed genes (i.e., on genes). Similarly, among 248 SHH-MB specific downregulated genes, 6 genes did not express in SHH-MB tumor cells specifically (*p* < 0.001, and expression in SHH-MB tumor cells < 1.5), referred to as SHH-MB tumor cell-specific non-expressed genes (i.e., off genes). Given the SHH-MB tumor cell-specific on and off genes, their expression in a bulk tissue profile is $g=g_{SHH\_on}^{ref}*purity+g_{SHH\_off}^{ref}*(1-purity$), with $g_{SHH\_on}^{ref}$ and $g_{SHH\_off}^{ref}$ as the reference expressions of the SHH-MB tumor cell-specific on and off genes, respectively.

### 2.5. Inferring miRNA Activity with ActMiR

We previously developed a tool for inferring miRNA activity, ActMiR [46,56], based on the expression levels of miRNA-predicted target genes in a tissue. Using ActMiR, we trained pediatric brain-specific miRNA activity models using the miRNA expression and gene expression data in GSE42658 [46,49]. The method is regression-based concerning both the miRNA expression and gene expression. In brief, the ActMiR method first determined the baseline expression levels of each miRNA’s target genes, i.e., a state where the miRNA did not regulate gene expression. This baseline expression level was defined as the average expression of samples with low miRNA expression. Next, we took the differences between expression levels of the target genes and the baseline expression level for each sample to determine how degraded the genes’ expressions were by miRNA. Finally, we performed a linear regression between these degradation values and the baseline expression; the resulting coefficient from the fit is the miRNA activity. However, because not all predicted miRNA target genes are functionally regulated by miRNAs, we determined the miRNA functional targets by using an iteratively reweighted least squares regression method between activity and gene expression [36]. High anticorrelation between miRNA activity and gene expression indicates that the gene is a functional target of miRNA. We inferred the activity and determined the functional target genes for the let-7 miRNA family using the mean miRNA expression of the following let-7 miRNA family members: hsa-let-7a, hsa-let-7b, hsa-let-7c, hsa-let-7d, hsa-let-7e, hsa-let-7f, hsa-let-7g, and hsa-let-7i.

Using the functional target genes determined by ActMiR in GSE42658, we inferred activity in the GSE85218 MB dataset [16] following the procedure, as previously described [56]. In brief, based on the expression levels of the negatively associated functional target genes of each miRNA, we calculated the sum of the scaled expression levels for each sample. We defined the baseline samples for each miRNA as the samples with the lowest sum of scaled expression levels of negatively associated functional target genes. We defined the bottom 5% of the total samples as the baseline samples. After defining the baseline samples, the procedure to estimate the miRNA activity is the same as the standard ActMiR procedure described above. As MB gene expression is heterogeneous by subgroup, we inferred the activity on each subgroup separately using the classifications defined in Reference [16].

### 2.6. Survival Association of miRNA Activity or Gene Expression 

We tested for the overall survival associations with miRNA activity. We used Cox proportional hazards regression on each MB subgroup with activity. We also separated samples within each subgroup based on whether their miRNA activity was higher or lower than the linear regression line between activity and tumor purity. We fit Kaplan–Meier survival curves and tested the equivalence of the curves using log-rank tests for high vs. low activity [57]. We performed the same procedure for the gene expression levels.

### 2.7. Interaction between Activity and Gene Expression

We investigated the relationship between the let-7 family’s activity and MYC, MYCN, and LIN28A/B, because they are in the same signaling pathway [35]. We calculated correlations between the gene expression and activity. We also separated samples based on subgroup-specific MYCN expression and whether the samples were greater or less than the purity-adjusted subgroup-specific expression (linear regression line between expression and tumor purity). The resulting four groups were high activity and high expression (MYCN^high^-let-7 activity^high^), high activity and low expression (MYCN^low^-let-7 activity^high^), low activity and high expression (MYCN^high^-let-7 activity^low^), and low activity and low expression (MYCN^low^-let-7 activity^low^).

### 2.8. Identification of Functional Target Genes Enriched for Canonical Pathways

We annotated the function of miRNAs by comparing their functional target genes with 1329 canonical pathways from MSigDB databases [58] and identified biological pathways overrepresented in the functional target gene set of each miRNA using Fisher’s exact test.

### 2.9. Detecting Small Molecules That Might Be Effective to SHH Subtypes with Poor Prognosis

First, we determined the differentially expressed genes (DEGs) between SHH-MB of high let-7 activity and high MYCN expression vs. the rest of SHH-MB by using *t*-tests. We determined DEGs at FDR 1%, which corresponded to *p*-values < 1.4 × 10^−4^. To calculate the FDR based on the *p*-value, the multipletests function in the statsmodels package in Python with the Benjamini and Hochberg method was used [54]. Next, based on differentially expressed genes, we determined whether these genes were upregulated or downregulated for tumors with high let-7 activity and high MYCN expression. Then, we investigated whether these genes were perturbed by drug treatments using the query tool (https://clue.io/query/, accessed on 3 June 2021) from the Connectivity Map (CMap) of the Library of Integrated Network-Based Cellular Signatures (LINCS) gene expression resource [59,60]. A negative enrichment score from CMap indicated that the treatment of the drug showed opposite expression changes for SHH-MB tumors with high let-7 activity and high MYCN expression compared to other SHH-MB tumors.

### 2.10. Validation

To validate our observations, we applied the analyses on 194 MB mRNA expression profiles generated by the St. Jude group in the following datasets [61]: GSE10327 [20], GSE12992 [62], and GSE30074 [63] and from http://www.stjuderesearch.org/data/medulloblastoma/, accessed on 26 March 2021 [28], which, overall, included 46 SHH-MB samples. We normalized these mRNA expression profiles to the GSE85218 SHH samples using ComBat in the sva R package [64]. We inferred the miRNA activity based on the models trained on the GSE42658 dataset. To assess the survival as sociations between let-7 activity and MYCN expression, we separated the samples into four groups based on purity-adjusted expression and let-7 activity (above or below the linear regression lines between let-7 activity/MYCN expression and tumor purity in GSE85218) and calculated the survival differences between different groups.

### 2.11. Computational Methods

Analysis was carried out using Python version 3.7.9 and R version 3.6.1 using the packages scipy, lifelines, and survival [65,66,67]. The figures were generated with seaborn and ggplot2 [68,69,70].

## 3. Results

### 3.1. Inter- and Intra-Tumor Heterogenity of MB Accessed by Cis-Methylation Regulated Genes

MB is categorized into four molecular subgroups according to gene expression patterns and clinical features [8,9,10,11]. The molecular subgroups of MB have been shown to be associated with distinct clinical features [8,9,10,11,12,13,14]. Here, we examined a MB cohort [16] with the multi-omics profiling data available (Methods). The subgroup and age characteristics of the MB cohort are described in Table 1. First, we used a PCA on the gene expression, and as expected, it was able to separate the MB samples into the four molecular subgroups [8,9,10,11,12,13,14] (Figure 1A). We further explored whether the expression of genes primarily regulated by methylation (Methods) can better cluster MB into subgroups, because DNA methylation shows a highly dynamic pattern during cellular differentiation, indicating its key function related to cell fate specification [71,72,73], and patterns of DNA methylation may provide an indirect assessment of these developmental origins. Indeed, cis-methyl genes (i.e., expression regulated by the methylation level in its promoter region; see Materials and Methods for details) can better separate MB into subgroups (Figure 1B) than all the genes (Figure 1A). In contrast, cis-CNV genes (i.e., expression regulated by its DNA copies; see Materials and Methods for details) did not improve the subgroup separation (Figure 1C,D).

This result indicates that the expression of cis-methyl genes can capture epigenetic fingerprints linked with the subgroups. Indeed, different molecular subtypes reflect different cells of origin of tumor cells [15,74], which have unique epigenetic fingerprints [29]. Furthermore, the expression of these cis-methyl genes that associate with cell types can be used to estimate the tumor cell purity in tumor samples. For example, the expression of cis-methyl genes can clearly separate SHH-MB (black dots in Figure 1B) from other MB subgroups. By integrating single-cell RNA sequencing (scRNAseq) data of the MB samples, we identified a set of cis-methyl genes whose expression was exclusively on or off in SHH-MB tumor cells (PN progenitor cells, [74]) (see Materials and Methods for details). As expected, the expression of this set of genes was able to separate SHH-MB from the other MB subgroups (Appendix A). Then, the SHH-MB tumor cell fraction in tumor samples was estimated based on this set of genes (see Materials and Methods).

In addition, the MB subgroups have varying amplifications/deletions in oncogenic genes [16,17]. As consistent with previous studies [16,17], both MYC and MYCN were cis-CNVs (*p* = 1.096 × 10^−8^ and 5.234 × 10^−11^, respectively; Materials and Methods). Furthermore, we found that MYC and MYCN are differently expressed by the subgroup (Figure 2A,B). The Group 3 and WNT subgroups showed higher MYC expression levels, while the SHH and WNT subgroups showed higher MYCN expression levels than the other two subgroups. The higher expression of MYCN in SHH compared to the other subgroups is consistent with known MYCN amplifications within the subgroup [16].

### 3.2. MYCN/LIN28/Let-7 Axis

MYCN, LIN28, and let-7 form a feed-forward loop (Figure 3), with MYCN promoting the transcription of LIN28A/B and LIN28A/B inhibiting the maturation of let-7 microRNAs and then let-7 inhibiting MYCN post-transcriptionally [35,37,38,75,76,77]. The MYCN/LIN28/let-7 axis plays a critical role in SHH-MB [16,17,26,27,28]. The expression of LIN28B had subgroup variations, while the expression of LIN28A did not (Figure 2C,D). The SHH and WNT subgroups had lower LIN28B expression levels than Groups 3 and 4 (Figure 2D), opposite of what would be expected based on the biology of high MYCN leading to high LIN28A/B expression (Figure 3), suggesting other molecular mechanisms regulating the feed-forward loop.

Next, we examined the overall survival associations of the expression of LIN28A, LIN28B, MYC, and MYCN using Cox proportional hazards regression. The expression of MYC, MYCN, and LIN28B was associated with the overall survival, as expected when examining all the MB samples together (Table 2), as their expression is also associated with the subgroups (Figure 2). When examining their associations with survival in each individual subgroup, the MYCN expression level was not associated with survival in SHH-MB (Table 2), even though MYCN was highly expressed in SHH-MB (Figure 2). On the other hand, the LIN28B expression level was low in SHH-MB (Figure 2), suggesting that other genes may play an important role in the MYCN/LIN28/let-7 pathways, such as let-7 miRNAs. Therefore, the let-7 miRNA family was further examined for their impact on the survival of SHH-MB.

### 3.3. MB Has Higher Let-7 Activity Than Other Brain Tissues and Subgroup-Specific Activity

Let-7 miRNAs are tumor suppressors [37,38,78] post-transcriptionally repressing MYCN expression, as well as feedback repressing LIN28 expression in the MYCN/LIN28/let-7 pathway [35,39,40]. Therefore, we explored the prognostic effect of let-7 activity in MB. There was no MB dataset with samples profiled for both miRNA and mRNA expression, so we examined miRNA profiles of 57 pediatric brain tumors (GSE42658) [49] that resembled MB. Within this dataset, the let-7 miRNAs had higher expression than the other miRNAs (Figure 4A), and the let-7 expression in MB was not significantly different from its expression level in the other tissues (MB vs. other tissues, *t*-test *p* = 0.65, Figure 4B).

Several studies have demonstrated that miRNA functional activity was not accurately reflected by its expression level [42,43,44], so we inferred the activity of let-7 using the previously described ActMiR method [46,56]. All let-7 miRNAs were highly expressed in pediatric brain tumors (Figure 4A), so we inferred let-7 activity using the mean expression of all let-7 miRNAs rather than individual miRNAs. Although let-7 miRNAs were not significantly differentially expressed among the different tissues (Figure 4B), let-7 activity in MB was significantly higher than the activity in most other tissues in a one vs. others comparison (*t*-test, *p* = 0.011), except the activity in choroid plexus papilloma (Figure 4C), indicating the potential regulatory role of let-7 miRNAs in MB.

The functional target genes of the let-7 miRNAs, estimated by integrating the miRNA expression and gene expression data of GSE42658, were used to infer the subgroup-specific let-7 activity for the 763 MB samples in GSE85218 using the ActMiR method [46] (see Materials and Methods for details). The subgroups SHH and Group 4 had 136 and 145 let-7 functional targets, respectively, much larger than the 33 and 38 functional targets in WNT and Group 3, respectively (Figure 5A). The SHH and Group 4 subgroups were enriched for a MYCN amplification [16], suggesting a potential relationship between MYCN and let-7 miRNA activity.

To annotate the function of miRNAs, we assessed the biological pathways over-represented in the functional target genes of the let-7 miRNAs (see Materials and Methods for details) [58] at 5% FDR corresponding to *p*-values < 1 × 10^−3^. The functional targets of let-7 in SHH-MB were uniquely significantly enriched for the Reactome G1 Phase pathway (Appendix A).

Next, because let-7 miRNAs suppress the MYCN expression post-transcriptionally [35] (Figure 3), we examined the correlation between let-7 activity and MYCN expression. In SHH-MB, let-7 activity had a significant negative correlation with MYCN expression (*p* = 1.66 × 10^−8^; Figure 5B), while let-7 activity was not significantly correlated with MYCN expression in the other subgroups (Figure 5C–E). This observation also indicates a potential regulatory role of let-7 miRNAs in the MYCN/LIN28/let-7 pathways in SHH-MB.

### 3.4. Stratifying SHH-MB by MYCN Expression and Let-7 Activity

The SHH-MB subgroup is the most heterogeneous among the MB subgroups [16,55]. As MYCN is highly expressed in SHH-MB (Figure 2) and let-7 miRNA is potentially active (Figure 5) in SHH-MB, we investigated the potential of MYCN expression and/or let-7 activity as a prognostic biomarker (Table 3). MYCN is amplified in some SHH-MB tumor cells [16,55], such that MYCN expression was significantly correlated with SHH-MB tumor purity, as expected (Spearman’s correlation coefficient = 0.3265, *p* = 6.193 × 10^−7^; Appendix A). On the other hand, let-7 activity was significantly anticorrelated with SHH-MB tumor purity (Spearman’s correlation coefficient = −0.3891, *p* = 1.788 × 10^−9^; Appendix A). Note that the LIN28A/B expression levels were very low (Figure 2 and Appendix A) and did not correlate with the tumor purity (Spearman’s correlation coefficient r = 0.02 and −0.01, *p* = 0.7 and 0.85 for LIN28A and LIN28B, respectively).

Next, the SHH-MB samples were partitioned into high/low MYCN expression groups based on the purity-adjusted mean MYCN expression level (Appendix A) or into high/low let-7 activity groups based on the purity-adjusted mean let-7 activity (Appendix A). Neither MYCN or let-7 were significantly associated with the survival rate (MYCN: log-rank test *p* = 0.182, Figure 6A and let-7: log-rank test *p* = 0.362, Figure 6B). When combining MYCN and let-7 activity information together, the SHH-MB patients with both high MYCN expression and high let-7 activity (MYCN^high^-let-7 activity^high^) had a much worse overall survival rate than the other three groups (MYCN^high^-let-7 activity^low^, MYCN^low^-let-7 activity^high^, and MYCN^low^-let-7 activity^low^) (log-rank test *p* = 0.0028, Figure 6C).

### 3.5. Validation in an Independent MB Cohort

To validate our observations, we combined 194 MB mRNA expression profiles generated by the St. Jude’s Children’s Research Hospital group [61], including GSE10327 [20], GSE12992 [62], GSE30074 [63], and from http://www.stjuderesearch.org/data/medulloblastoma/, accessed on 26 March 2021 [28] (see Materials and Methods), referred to as the validation cohort collectively. The characteristics of the validation cohort are summarized in Table 1. Like the GSE85218 dataset, the MYCN expression in SHH-MB was significantly higher than in the other groups (*p* = 1.420 × 10^−9^, Appendix A). The same analyses above were applied to the validation cohort (the results are listed in Table 4).

For SHH-MB in the validation cohort, the inferred let-7 activity and MYCN expression were significantly anticorrelated (Spearman’s correlation r = −0.393, *p* = 0.00689) (Appendix A), consistent with the observations in the GSE85218 data. Their associations with survival are listed in Table 5. Similarly, the inferred SHH-MB tumor purity was correlated with MYCN expression (r = 0.2961, *p* = 0.04572) and anticorrelated with let-7 activity (r = −0.3261, *p* = 0.0270). Then, the SHH-MB samples were partitioned into high/low MYCN expression or high/low let-7 activity groups based on the purity-adjusted mean values (Appendix A). MYCN expression or let-7 activity alone were not significantly associated with survival (*p*-values = 0.27 and 0.63, respectively; Figure 7A,B). When combining MYCN expression and let-7 activity information together, the SHH-MB patients with both high MYCN expression and high let-7 activity had a significantly worse survival rate than MYCN^high^-let-7 activity^low^, MYCN^low^-let-7 activity^high^, and MYCN^low^-let-7 activity^low^ (log-rank test *p*-value = 0.037; Figure 7C). All these findings were consistent with our observations in the GSE85218 dataset.

### 3.6. Drug Repurposing to Identify Potential Therapeutic Treatment for SHH Subgroups

MYCN-amplified SHH-MB is generally resistant to SMO inhibitors, a targeted therapy for SHH-MB [18,19,79]. It has been suggested that DFMO would have a greater effect on MYCN-amplified tumors [80]. ODC inhibition by DFMO decreases LIN28 and increases let-7 [35,76,81], so the low expression of LIN28A/B in SHH-MB and the high let-7 activity suggest that DFMO is unlikely to be effective for SHH-MB patients with high MYCN expression and high let-7 activity who have a less-favorable survival rate. Thus, there is an urgent need to develop therapeutics for SHH-MB patients, especially for patients with high let-7 activity and high MYCN expression.

To understand the molecular mechanisms driving the survival differences (Figure 6), we compared SHH-MB with both high MYCN expression and high let-7 activity against the other SHH-MB samples and identified 172 differentially expressed genes at FDR < 1%, including 125 upregulated and 47 downregulated genes in the MYCN^high^-let-7 activity^high^ SHH-MB group. Using these genes, we performed a pathway enrichment analysis and found that the upregulated genes were enriched for KEGG neuroactive ligand receptor interaction (*p* = 3.448 × 10^−5^) and other ligand–receptor pathways, which may indicate cell-to-cell interactions as potential drug targets (Appendix A).

Next, we compared these differentially expressed genes with drug signatures in the Connectivity Map (CMap) of the Library of Integrated Network-Based Cellular Signatures (LINCS) gene expression resource [59,60] using the query tool (https://clue.io/query/, accessed on 3 June 2021). We identified 21 drug candidates whose effects may reverse the DEG signature (negative scores lower than −90, Table 6). Some drugs with the same mechanisms of actions showed consistent enrichment scores < −90. For example, FGFR inhibitors, including orantinib, PD-173074, and brivanib, were in the candidate drug list (Table 6). Previous studies have shown that FGFR promotes MB tumor cell invasion in vitro [82], the blockade of FGFR represses brain tissue infiltration in vivo [82], and FGFR inhibitor decreases the viability and proliferation in MB cell lines [83]. These results suggest that the FGFR signaling pathway may be a potential target for treating SHH-MB patients with both high MYCN expression and high let-7 activity.

## 4. Discussion

Our integrative analysis of gene expression, methylation, and CNV identified cis-methylation genes that better separated MB patients into molecular subgroups than all genes, indicating subgroups are probably linked to an epigenetic memory of development lineages [15,29,55,74]. By leveraging inferred miRNA activity, we identified a subset of SHH-MB patients with a worse survival rate than other SHH-MB patients (Figure 6C), and the association was validated in an independent SHH-MB cohort (Figure 7C). The MYCN^high^-let-7 activity^high^ SHH-MB subset did not overlap with the SHH-MB subtypes based on a similarity network fusion (SNF) analysis [16]. SHH-α and SHH-β, subtypes of SHH-MB by SNF, were enriched for MYCN amplification and were associated with worse survival outcomes [16] (five-year survival 69.8% and 67.3% for SHH-α and SHH-β, respectively vs. 88–88.5% for other SHH-MBs). The MYCN^high^-let-7 activity^high^ SHH-MB patient group consisted of 32 samples, where 10 of the samples fell into the SNF subtypes SHH-α or SHH-β (five and five samples, respectively) (Fisher’s exact test *p* = 0.05 and *p* = 0.6).

High MYCN expression being associated with inferior survival in SHH-MB is expected, as MYCN is an oncogene [16]. However, let-7, a known tumor suppressor, is downregulated in multiple tumor types [37,78,84]. It is counterintuitive that MYCN^high^-let-7 activity^high^ SHH-MB patients had a worse overall survival rate than the other SHH-MB patients, especially MYCN^high^-let-7 activity^low^ SHH-MB patients. It has been shown that high levels of MYCN protein persist in MYCN-amplified neuroblastoma after transfecting high levels of let-7 miRNA and that MYCN, which is the most abundant target for let-7 miRNA, actually acts as a sponge for let-7 miRNAs in these cells [85]. Our observation that MYCN^high^-let-7 activity^high^ SHH-MB was associated with worse survival might be due to the MYCN sponge effect on let-7 miRNAs dominating the let-7-repressing effect on MYCN transcripts. Indeed, MYCN expression and let-7 activity were not correlated in MYCN^high^-let-7 activity^high^ SHH-MB, while they were significantly anticorrelated in other SHH-MB (Appendix A).

High LIN28 expression is known to be significantly correlated with a shorter survival time in MB [86]. LIN28B regulates multiple oncogenic processes, in part by downregulating the let-7 miRNA family in MB, and LIN28B expression is associated with survival in Group 3 and Group 4 MB [29]. However, whether LIN28/let-7 play an important role in SHH-MB is not previously known, indicating that our finding that let-7 plays a potential critical regulatory role in SHH-MB is novel. Beyond validation in independent datasets (Figure 7), experimental validations, such as RT-qPCR after let-7 knockdown or overexpression in SHH-MB cells, are needed to confirm the regulatory relationships between let-7 and its target genes.

About 8% of SHH-MB patients carry TP53 mutations based on a whole-genome sequencing study [12]. A separate study showed that SHH-MB with TP53 mutations have worse survival than SHH-MB without TP53 mutations [23]. The functional impact of TP53 mutations is heterogeneous, resulting in a spectrum of functional changes from loss of function to gain of function [87]. In our training dataset (GSE85218), the TP53 mutation status of five SHH-MB tumor samples was characterized by whole-genome sequencing [12,88]. One of the five SHH-MB samples carried a TP53 mutation. To assess the relationship between the TP53 mutation status and potential p53 protein function, we estimated the p53 pathway activity for each SHH-MB sample in the training dataset based on ssGSEA [89,90]. The SHH-MB sample with TP53 mutations was within the lowest quartile of the p53 pathway activity (Appendix A); however, SHH-MB patients with p53 pathway activity lower than the activity in this TP53-mutated patient had better survival than the other SHH-MB patients (Appendix A). These results together indicate a high complexity in including the TP53 mutation status or p53 functional status in prognostic biomarker models.

The overall survival rates of the four MB subgroups were not significantly different (Appendix A), and SHH-MB was the most heterogeneous subgroup among the four MB subgroups [12]. SMO inhibition is a targeted therapy for SHH-MB, but SHH-MB patients with high MYCN expression were resistant to SMO inhibitors in general [18,19,79]. Furthermore, p53 mutations are associated with MYCN amplification and are implicated in conferring resistance to both radiation therapy and SMO inhibitors in SHH-MB [22,23,24,25]. A subset of SHH-MB patients with high MYCN expression (MYCN^high^-let-7 activity^high^ SHH-MB) had a significantly worse survival rate than the other SHH-MB (Figure 6 and Figure 7). Thus, it is urgent to identify effective treatments for these MYCN^high^-let-7 activity^high^ SHH-MB patients. DFMO, an inhibitor of LIN28A/B, has been identified as a potential treatment option for embryonal tumors [80]. It is worth noting that the LIN28A/B expression levels were low in SHH-MB (Figure 2) and similar to other tissues (Appendix A), so it is unlikely that DFMO is effective in SHH-MB. Our analysis suggests that FGFR inhibitors are potential drug candidates for MYCN^high^-let-7 activity^high^ SHH-MB (Table 6). We note that the effects of these predicted candidates have not been validated in SHH-MB cancers. Further in vitro and in vivo experiments are needed to validate and strengthen our findings and demonstrate their therapeutic values. Our results suggest that the MB patients in each molecular subtype are still heterogeneous, and integrated genomic analyses can be used for dissecting their heterogeneity and identifying novel subtype-specific prognostic biomarkers and therapeutic targets.

## 5. Conclusions

We applied an integrated genomic analysis on MB datasets and inferred the let-7 miRNA activity in MB. We identified a SHH-MB subset with high MYCN expression and high let-7 activity associated with a worse survival rate than the other SHH-MB and validated the association in an independent MB cohort.

## Figures and Tables

**Figure 1 cancers-14-00139-f001:**
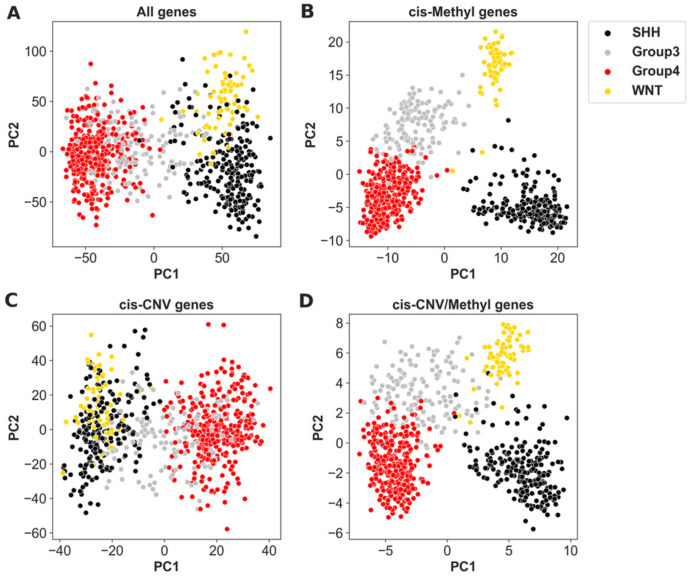
Subgroup distributions based on the PCA of expression profiles of (**A**) all 21,050 genes measured within GSE85212, GSE85217, and GSE37384; and (**B**) the 589 cis-methyl genes; (**C**) 3630 cis-CNV genes; and (**D**) 107 genes in the overlap of the cis-CNV and cis-Methyl genes.

**Figure 2 cancers-14-00139-f002:**
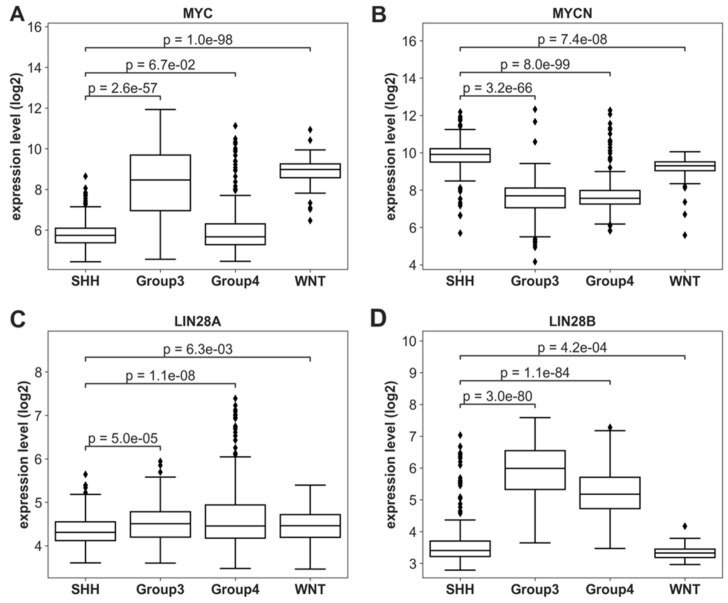
Expression profiles for critical genes in the LIN28 pathway. (**A**–**D**) The expression distributions of MYC, MYCN, LIN28A, and LIN28B, respectively, in GSE85218. LIN28A has a similar expression for all four subgroups. SHH and WNT have lower expressions than Group 3 and Group 4 in LIN28B and higher expression in MYCN [16,20]. MYC has a unique expression profile: SHH and Group 4 have lower expressions, while WNT and Group 3 have higher expressions. ◊: Diamonds indicate outliers.

**Figure 3 cancers-14-00139-f003:**
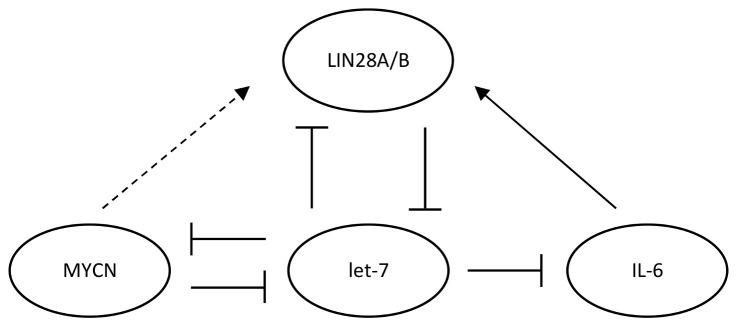
LIN28 signaling pathway. MYC, MYCN, and IL-6 promote transcription for LIN28A/B. let-7 miRNAs post-transcriptionally repress the expression of MYC, MYCN, and IL-6.

**Figure 4 cancers-14-00139-f004:**
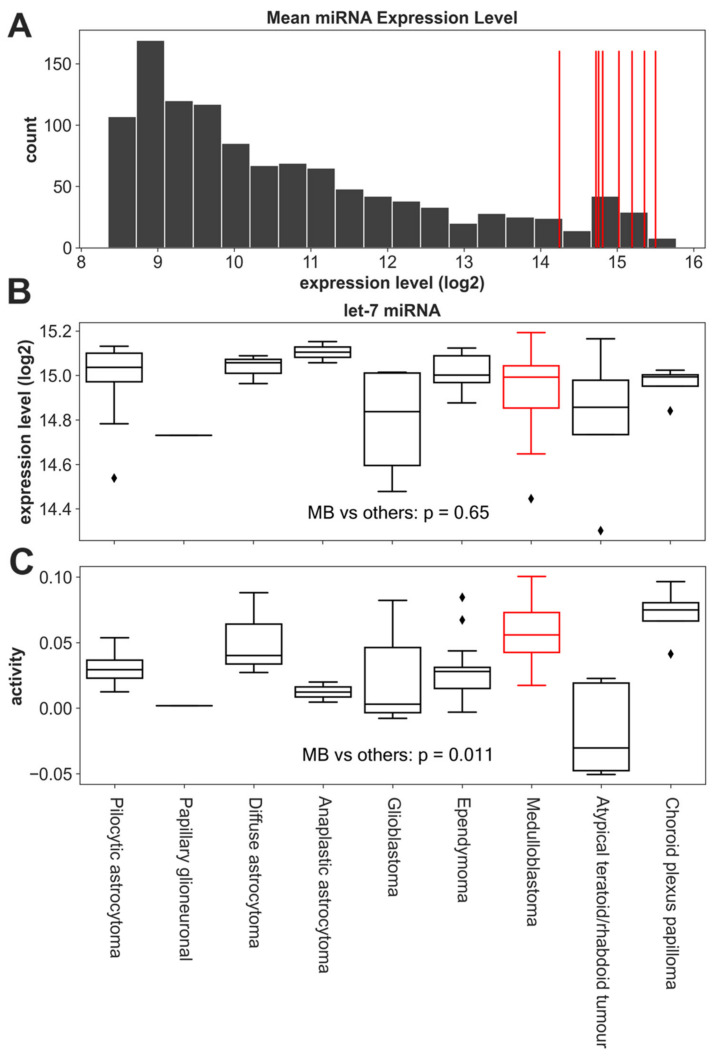
Distributions of the miRNA expression in GSE42658. (**A**) A histogram of the mean expression levels of all the miRNAs. Let-7 family miRNAs are indicated as red vertical lines. (**B**) Boxplots of the mean let-7 family expression across tissue types. The expression of let-7 members in MB (red) is not significantly different from its expression in other tissues or tumor types. (**C**) The inferred let-7 family activity across tissues. The Let-7 activity in MB (red) was significantly higher than in the other tissues in a one vs. others comparison (*t*-test, *p* = 0.011). A higher activity of let-7 in MB with similar expression levels to other tissues indicates that the let-7 family more frequently regulates the expression of its target genes in MB compared to other tissues [46,56]. ◊: Diamonds indicate outliers.

**Figure 5 cancers-14-00139-f005:**
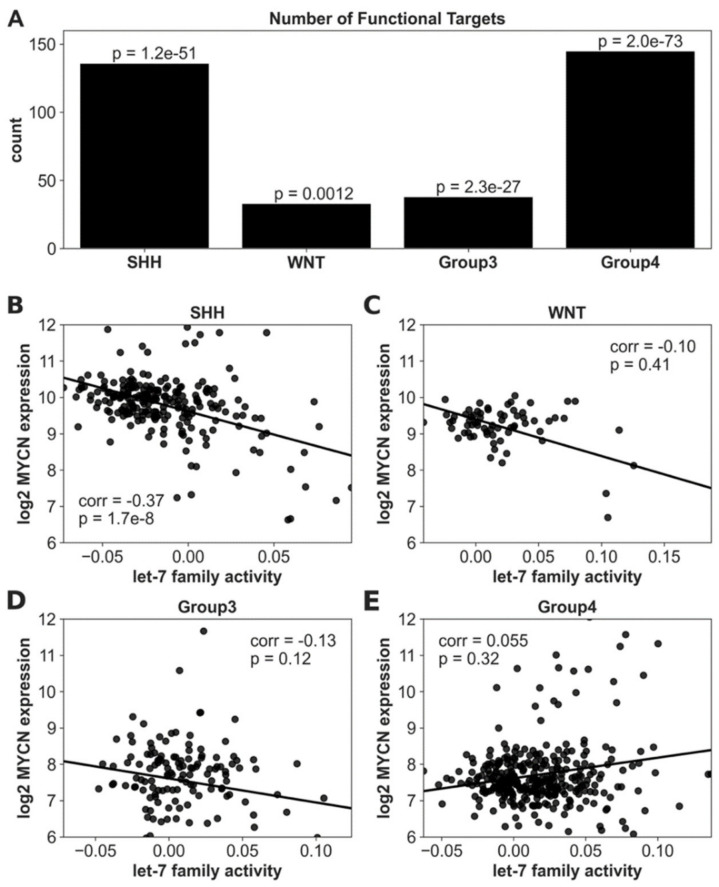
Inferred activity in the let-7 miRNA family and correlation against MYCN for samples in GSE85218. (**A**) The bar plot of the number of inferred functional targets for the let-7 family. *p*-values indicating enrichment for functional targets by the hypergeometric test. SHH and Group 4 have the most functional targets of the let-7 family. (**B**–**E**) Correlations of the let-7 activity and MYCN expression in the subgroups SHH, WNT, Group 3, and Group 4, respectively. In SHH-MB, let-7 activity and MYCN expression had a significant Spearman’s correlation, with *p* = 1.66 × 10^−8^. The correlation *p*-values were 0.411, 0.117, and 0.319 in WNT, Group 3, and Group 4, respectively.

**Figure 6 cancers-14-00139-f006:**
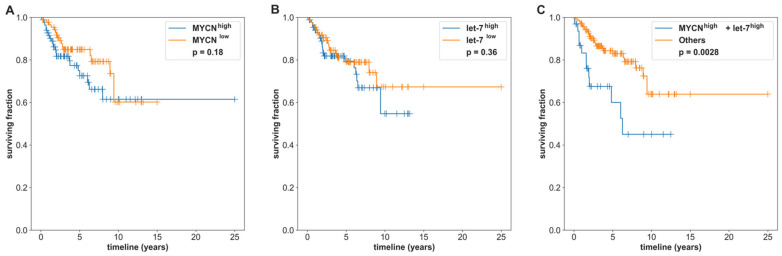
Kaplan–Meier overall survival curves for SHH-MB based on the MYCN expression, let-7 family activity, and MYCN + let-7 family activity for the samples in GSE85218. High activity/expression (blue) was defined as the group of samples with activity/expression greater than or equal to the linear regression line between the MYCN/let-7 family and purity, and low activity/expression (orange) was defined as the group of samples with activity/expression less than the linear regression line. (**A**) The Kaplan–Meier survival curves for SHH with high vs. low MYCN expression (log-rank test *p* = 0.182), (**B**) the Kaplan–Meier survival curves for SHH with let-7 activity (*p* = 0.362), and (**C**) the Kaplan–Meier survival curves for let-7 activity and MYCN expression. MYCN^high^-let-7 activity^high^ showed a significantly worse outcome (log-rank tests with *p* = 0.0028) than the combined group (Others) of MYCN^high^-let-7 activity^low^, MYCN^low^-let-7 activity^high^, and MYCN^low^-let-7 activity^low^. The Others group in 7C is represented as individual groups in Appendix A.

**Figure 7 cancers-14-00139-f007:**
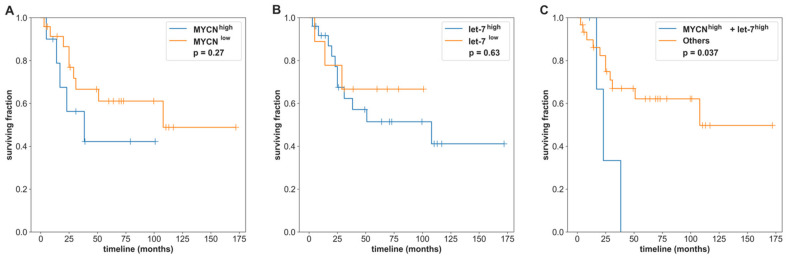
Associations of the survival rate and MYCN expression and let-7 activity based on SHH-MB samples in the validation datasets. High activity/expression (blue) was defined as the group of samples with activity/expression greater than or equal to the linear regression line between MYCN/let-7 family and purity, and low activity/expression (orange) was defined as the group of samples with activity/expression under the linear regression line. (**A**) The Kaplan–Meier overall survival curves for MYCN expression (log-rank test *p* = 0.267), (**B**) the Kaplan–Meier overall survival curves for let-7 family activity (log-rank test *p* = 0.627), and (**C**) the Kaplan–Meier survival curves for let-7 activity with MYCN expression. MYCN^high^-let-7 activity^high^ (blue) showed a significantly worse outcome (log-rank tests *p* = 0.037) than MYCN^high^-let-7 activity^low^, MYCN^low^-let-7 activity^high^, and MYCN^low^-let-7 activity^low^.

**Table 1 cancers-14-00139-t001:** Number of samples by subgroup and age for the studies, as described by Northcott et al. [16]. The validation datasets had incomplete age information [20,28,61,62,63].

Dataset	Subgroup	Subgroup Number	Age	Number
Training (GSE85218)	SHH	223	0–3	62
			4–10	55
			10–17	29
			18+	69
	WNT	70	0–3	1
			4–10	23
			10–17	27
			18+	13
	Group 3	144	0–3	24
			4–10	90
			10–17	17
			18+	5
	Group 4	326	0–3	11
			4–10	181
			10–17	108
			18+	14
Validation	SHH	46	<3	23
			≥3	22
	WNT	21	<3	0
			≥3	21
	Group 3	37	<3	9
			≥3	27
	Group 4	74	<3	3
			≥3	71

**Table 2 cancers-14-00139-t002:** Associations of the overall survival (OS) and genes in the LIN28 pathway. Using all MB samples together, LIN28B, MYC, and MYCN were significantly associated with survival. For SHH-MB, only LIN28B was significantly associated with the OS. In WNT-MB, none of the four genes was significant. In Group 3, MYC and MYCN were significant. In Group 4, MYC’s association was significant.

Subgroup	Gene	Hazard Ratio	Wald *p*	Log-Rank *p*
Training All Samples	MYCN	0.86	1.9 × 10^−2^	0.019
	MYC	1.1	4.9 × 10^−3^	4.6 × 10^−3^
	LIN28A	1.0	0.98	0.98
	LIN28B	1.4	1.1 × 10^−6^	8.0 × 10^−7^
Training SHH	MYCN	1.3	0.25	0.26
	MYC	0.97	0.91	0.91
	LIN28A	1.1	0.79	0.79
	LIN28B	1.5	0.01	9.2 × 10^−3^
Training WNT	MYCN	2.8	0.45	0.43
	MYC	0.40	0.26	0.27
	LIN28A	1.8	0.70	0.70
	LIN28B	0.82	0.92	0.92
Training Group 3	MYCN	0.74	0.028	0.029
	MYC	1.2	0.020	0.019
	LIN28A	0.71	0.29	0.29
	LIN28B	1.3	0.11	0.11
Training Group 4	MYCN	1.1	0.49	0.49
	MYC	1.3	1.01 × 10^−3^	8.4 × 10^−4^
	LIN28A	0.96	0.83	0.83
	LIN28B	0.97	0.87	0.87

**Table 3 cancers-14-00139-t003:** Associations of the overall survival rate and genes in the LIN28 pathway with let-7 activity based on the SHH-MB samples in GSE85218. LIN28A/B KM curves for SHH-MB are shown in Appendix A.

Test	Variable	Hazard Ratio	Wald *p*	Log-Rank *p*
Surv~MYCN	MYCN	1.3	0.25	0.26
Surv~LIN28A	LIN28A	1.1	0.79	0.79
Surv~LIN28B	LIN28B	1.5	0.011	0.0092
Surv~let-7	let-7	169.5	0.32	0.32
Surv~purity	purity	3.3	0.11	0.11
Surv~age	age	0.99	0.55	0.55
Surv~MYCN + let-7	MYCN	1.7	0.066	0.069
	let-7	2.9 × 10^5^		
Surv~purity + MYCN + let-7	purity	1.4 × 10^−3^	0.025	0.024
	MYCN	1.7		
	let-7	7.7 × 10^3^		

**Table 4 cancers-14-00139-t004:** Associations of the survival rate and genes in the LIN28 pathway based on samples in the validation dataset.

Subgroup	Gene	Hazard Ratio	Wald *p*	Log-Rank *p*
Validation all	MYCN	1.3	0.019	0.019
	MYC	1.0	0.58	0.58
	LIN28A	1.7	0.041	0.044
Validation SHH	MYCN	1.4	0.14	0.14
	MYC	0.77	0.49	0.49
	LIN28A	5.3	0.074	3.5 × 10^−5^
Validation WNT	MYCN	5.2	0.15	0.076
	MYC	0.20	0.23	0.17
	LIN28A	65.8	0.27	0.25
Validation Group3	MYCN	0.90	0.82	0.82
	MYC	1.2	0.30	0.29
	LIN28A	2.0	0.28	0.28
Validation Group4	MYCN	1.7	1.1 × 10^−3^	4.3 × 10^−4^
	MYC	1.1	0.64	0.64
	LIN28A	0.66	0.39	0.39

**Table 5 cancers-14-00139-t005:** Associations of the survival rate and genes in the LIN28 pathway, let-7 activity, and their combined effects for the SHH-MB samples in the validation dataset. LIN28A KM curves for SHH-MB are shown in Appendix A.

Test	Variable	Hazard Ratio	Wald *p*	Log-Rank *p*
Surv~MYCN	MYCN	1.4	0.14	0.14
Surv~LIN28A	LIN28A	5.3	0.074	3.5 × 10^−5^
Surv~LIN28B	LIN28B	N/A	N/A	N/A
Surv~let-7	let-7	11.4	0.55	0.55
Surv~purity	purity	0.20	0.81	0.81
Surv~age	age <3	1.2	0.79	0.79
	age ≥3	N/A		
Surv~MYCN + let-7	MYCN	1.8	0.039	0.025
	let-7	7.7 × 10^3^		
Surv~purity + MYCN + let-7	purity	0.98	0.090	0.057
	MYCN	1.8		
	let-7	7.7 × 10^3^		

**Table 6 cancers-14-00139-t006:** CMap results for the 21 drugs with negative CMap connectivity scores. The target genes of these drugs are shown as a full table in Appendix A. ^1^ CMap connectivity score. A score of 95 indicates that only 5% of the reference gene sets showed stronger connectivity than the current query to the perturbagen in question. ^2^ Number of drugs that have the same mechanism of action and of which scores are smaller than −90. The number in parentheses indicates the total number of drugs with the same mechanisms of action. ^3^ Enrichment statistics of drugs with the same mechanisms of action. Only shown scores are smaller than −90.

Score ^1^	Name	Description	Number of Drugs ^2^	Enrichment Score ^3^
−99.26	cobalt(II)-chloride	HSP inducer	1 (3)	-
−98.8	amonafide	Topoisomerase inhibitor	1 (16)	-
−98.64	embelin	HCV inhibitor	1 (3)	-
−97.22	hyperforin	Cyclooxygenase inhibitor	1 (57)	-
−97.02	parthenolide	NF-kB pathway inhibitor	1 (12)	-
−95.75	dapsone	Bacterial antifolate	1 (3)	-
−95.6	brivanib	FGFR inhibitor	3 (3)	−97.12
−94.79	ketoconazole	Sterol demethylase inhibitor	1 (6)	-
−94.68	piperacillin	Bacterial cell wall synthesis inhibitor	1 (29)	-
−94.61	ziprasidone	Dopamine receptor antagonist	1 (65)	-
−92.54	tienilic-acid	Sodium/potassium/chloride transporter inhibitor	1 (4)	-
−92.07	sitagliptin	Dipeptidyl peptidase inhibitor	1 (3)	-
−91.88	orantinib	FGFR inhibitor	3 (3)	−97.12
−91.59	FCCP	Mitochondrial oxidative phosphorylation uncoupler	1 (2)	-
−91.52	XMD-885	Leucine rich repeat kinase inhibitor	1 (3)	-
−91.3	geldanamycin	HSP inhibitor	1 (13)	-
−90.67	harpagoside	Acetylcholinesterase inhibitor	1 (17)	-
−90.59	tyrphostin-AG-1478	EGFR inhibitor	1 (42)	-
−90.43	AG-370	PDGFR receptor inhibitor	1 (9)	-
−90.06	PD-173074	FGFR inhibitor	3 (3)	−97.12
−90.05	kinetin-riboside	Apoptosis stimulant	1 (10)	-

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
