# Peer review of "Identification of Let-7 miRNA Activity as a Prognostic Biomarker of SHH Medulloblastoma"

_cancers, 2021, doi:10.3390/cancers14010139_

Round 1

Reviewer 1 Report

In the present manuscript "Identification of let-7 miRNA activity as a prognostic biomarker of SHH medulloblastoma" the authors have performed an extensive in silico analysis of MYCN and let-7 family clinical value in Medulloblastoma (MB). The manuscript is well written and the findings support in general the conclusions of the study. The quality/impact of the work can be improved by addressing the following comments.

  1. The overall presentation of the data (figures and tables) is average. In most cases the figures is really difficult to see/understand. The authors should prepare the figures in a more friendly way for the readers, including larger fonts, lager images etc.
  2. p-values should be included in figures
  3. Figure 2 should be replaced by boxplots including p-values
  4. Figures 4 and Figure 5, please include p-values
  5. Figure 6. I strongly believe that Fig. 6A and 6B should be presented in supplemental data as the statistical p are not significant. Moreover, in K-M plot of 6C please include only to curves, MYCN and let-7 high vs other. This would be more easy for the readers.
  6. The same for figure 7.
  7. Tables. Please include number up to 3 decimals.
  8. Experimental validation by targeted analysis (RT-qPCR) instead of high-throughput data will strengthen the study and findings
  9. In the discussion section the authors should indicate the limitations of the study, and most importantly the need for experimental validation by targeted analysis (RT-qPCR) instead of high-throughput data.
  10. The authors should discuss further the reason for MYCN/let-7 vclinical value in SHH group vs the other molecular groups of MB.   

Author Response

In the present manuscript "Identification of let-7 miRNA activity as a prognostic biomarker of SHH medulloblastoma" the authors have performed an extensive in silico analysis of MYCN and let-7 family clinical value in Medulloblastoma (MB). The manuscript is well written and the findings support in general the conclusions of the study. The quality/impact of the work can be improved by addressing the following comments.

We thank the reviewer for encouraging comments and detailed suggestions.

  1. The overall presentation of the data (figures and tables) is average. In most cases the figures is really difficult to see/understand. The authors should prepare the figures in a more friendly way for the readers, including larger fonts, lager images etc.

We modified all figures for better visual effects.

  1. p-values should be included in figures

We updated and included p-values in Figs 2, 4-7.

  1. Figure 2 should be replaced by boxplots including p-values

We thank the reviewer for the suggestions.  We updated Figure 2 as suggested.

  1. Figures 4 and Figure 5, please include p-values

We updated Figures 4 and 5 as suggested.

  1. Figure 6. I strongly believe that Fig. 6A and 6B should be presented in supplemental data as the statistical p are not significant. Moreover, in K-M plot of 6C please include only to curves, MYCN and let-7 high vs other. This would be more easy for the readers.

We thank the reviewer for the suggestions.  We have updated Figure 6C as suggested and move the original Fig. 6C as Supplementary Figure 12. However, we think panels Figs 6A&B highlights that simple and straight-forward approaches do not work well. They are useful in the same figure as the panel 6C so that the difference can be easily seen.

  1. The same for figure 7.

We have updated Figure 7C as suggested and have moved the original Fig. 7C to Supplementary Figure 6.

  1. Tables. Please include number up to 3 decimals.

We have updated tables as the reviewer suggested.

  1. Experimental validation by targeted analysis (RT-qPCR) instead of high-throughput data will strengthen the study and findings

We appreciate the reviewer’s suggestion. Unfortunately, we don’t have access to the tissue samples from which multi-omics data were generated and analyzed in the study.  

  1. In the discussion section the authors should indicate the limitations of the study, and most importantly the need for experimental validation by targeted analysis (RT-qPCR) instead of high-throughput data.

We added the following paragraph in the Discussion section as the reviewer suggested:

“High LIN28 expression is known to be significantly correlated with shorter survival time in MB [87]. LIN28B regulates multiple oncogenic processes, in part by downregulating the let-7 miRNA family in MB and LIN28B expression is associated with survival in Group 3 and Group 4 MB [29]. However, whether LIN28/let-7 plays an important role in SHH-MB is not previously known, indicating our finding that let-7 plays a potential critical regulatory role in SHH-MB is novel. Beyond validation in independent datasets (Fig. 7), experimental validations, such as RT-qPCR after let-7 knockdown or overexpression in SHH-MB cells, are needed to confirm regulatory relationships between let-7 and its target genes.”

  1. The authors should discuss further the reason for MYCN/let-7 vclinical value in SHH group vs the other molecular groups of MB.   

We revised the following paragraph in the Discussion section as the reviewer suggested: “Overall survival rates of the four MB subgroups were not significantly different (Suppl. Fig. 10), and SHH-MB is the most heterogeneous subgroup among the four MB subgroups [12]. SMO inhibition is a targeted therapy for SHH-MB, but SHH-MB patients with high MYCN expression were resistant to SMO inhibitors in general [18,19,80]. Furthermore, p53 mutations are associated with MYCN amplifications, and are implicated in conferring resistance to both radiation therapy and SMO inhibitors in SHH-MB [22–25]. A subset of SHH-MB patients with high MYCN expression (MYCNhigh-let-7 activityhigh SHH-MB) had significantly worse survival than other SHH-MB (Fig. 6&7). Thus, it is urgent to identify effective treatments for these MYCNhigh-let-7 activityhigh SHH-MB patients.”

Reviewer 2 Report

Westphal et al have used various bioinformatics approaches, including the ActMir platform, to identify high MYCN expression concurrent with high let-7 activity as a prognostic biomarker of sonic hedgehog (SHH) medulloblastoma (MB). Overall, the study is well done, but its overall significance to the field of paediatric neuro-oncology will require testing in a prospective clinical trial. Moreover, assessment of the role of FGFR inhibitors will require testing in clinical trials as well.

Major concern:

The authors have not incorporated p53 mutation into their risk classification nor discussed the role of p53 as an independent poor prognosis risk factor in earlier studies.

Specific concerns:

  1. Figure 4, Line 323: This statement is not correct based upon the presented data. CPP mean expression is higher than MB as per the box plots. Please revise.
  2. What was the rationale to exclude FGFR inhibition from the abstract? Was this due to the lack of functional validation in vitro?

Minor concerns:

Line 8: Spell "Hospital" correctly.

Line 13: Add "malignant" prior to "pediatric". The most common pediatric brain tumours are the low grade gliomas.

Line 21: Add "brain" after "embryonal".

Line 25: Replace "and" with "with" since it is the concurrent "high/high" state that is prognostic.

Line 56: Replace "were" with "are".

Lines 70, 73 and 74: Change "investigate" to "investigated".

Line 96: Of the 9 MB samples, how many were SHH? WNT? Group 3? Group 4?

Line 98: Did you mean "papillary"?

Line 193: Add "and" prior to "identified".

Line 198: Add "of" after "rest".

Line 199: Add "a" prior to "p-value".

Line 201: Add "the" after "with".

Line 232: Replace "is" with "was".

Line 307: The sentence is unclear as written. What is meant by "not significantly associated with any particular brain tissue"?

Line 320: Spell the second use of "let-7" correctly.

Line 384: Change to "St. Jude Children's Research Hospital" group.

Line 386: Replace "is" with "a"

Line 392: Rewrite as "observations in the GSE85218 data".

Lines 464-465: This sentence is very confusing as written: "worse survival outcomes identified with worse survival".

Line 499: Revise to "validated".

Author Response

Westphal et al have used various bioinformatics approaches, including the ActMir platform, to identify high MYCN expression concurrent with high let-7 activity as a prognostic biomarker of sonic hedgehog (SHH) medulloblastoma (MB). Overall, the study is well done, but its overall significance to the field of paediatric neuro-oncology will require testing in a prospective clinical trial. Moreover, assessment of the role of FGFR inhibitors will require testing in clinical trials as well.

We thank the reviewer for encouraging comments.  We completely agree with the reviewer that there is a long way to go from discovery to translational use.

Major concern:

The authors have not incorporated p53 mutation into their risk classification nor discussed the role of p53 as an independent poor prognosis risk factor in earlier studies.

We thank the reviewer for bring up this import issue on p53 mutations.  About 8% (10 out of 131) of SHH-MB carry TP53 mutations based on whole-genome sequencing study of 491 MB samples (Northcott et al, Nature, 2017).  An early study by Zhukova et al (J Clin Oncol, 2013) shows that SHH-MB with TP53 mutations have worse survival than SHH-MB without TP53 mutation.  However, the function impact of TP53 mutations is heterogeneous, resulting in a spectrum of functional changes from loss of function to gain of function (see review by Monti et al, Frontiers in Oncology, 2020).  In our training dataset (GSE85218), 26 SHH-MB tumor samples were further characterized by whole-genome sequencing (Skowron et al, Nature Commun., 2021). Among them, 5 samples’ mutation statuses are available for our analysis. One of the 5 SHH-MB samples carries TP53 mutation.  To assess the relationship between TP53 mutation status and potential p53 protein function, we estimated p53 pathway activity for each SHH-MB sample in the training dataset based on single sample GSEA (ssGSEA).  The sample with TP53 mutation was within lowest quartile of p53 pathway activity (Supplementary Figure 9A).  However, SHH-MB patients with p53 pathway activity lower than the activity in the TP53 mutated patient had better survival than the other SHH-MB patients (Supplementary Figure 9B).  These results together indicate high complexity in including TP53 mutation status or p53 function status in prognostic biomarker models.  We added a few sentences on TP53 in the Introduction and Discussion sections.

Figure S9.  TP53 in SHH-MB samples of GSE85218.  A)  the distribution of p53 pathway activity in SHH-MB. We applied single sample gene set enrichment analysis (ssGSEA) method in GSVA package [90] by using genes within Hallmark p53 pathways [91]. The red star indicates the sample with known TP53 mutation status while gray circles indicate without TP53 mutation. The TP53 mutation information is based on Northcott et al. [12] data set. The table indicating matched samples between GSE85218 and Northcott et al data was downloaded from Skowron et al. [89]. B) Kaplan-Meier survival curves for samples with p53 pathway activity scores lower than or equal to that of the sample with TP53 mutation (shown in blue) and the rest of samples (shown in red). 

Specific concerns:

  1. Figure 4, Line 323: This statement is not correct based upon the presented data. CPP mean expression is higher than MB as per the box plots. Please revise.

We have revised the sentence as “ let-7 activity in MB was significantly higher than the activity in most  other tissues in a one vs others comparison (t-test, p = 0.011) except the activity in choroid plexus papilloma (Fig. 4C), indicating the potential regulatory role of let-7 miRNAs in MB. ”

  1. What was the rationale to exclude FGFR inhibition from the abstract? Was this due to the lack of functional validation in vitro?

We thank the reviewer for the suggestion.  We included the following sentence in the abstract: “Comparing the expression differences between the two SHH-MB prognostic subtypes with compound perturbation profiles we identify FGFR inhibitors as one potential treatment option for SHH-MB patients with the less-favorable prognostic subtype.”

Minor concerns:

Line 8: Spell "Hospital" correctly.

We corrected the misspelling.

Line 13: Add "malignant" prior to "pediatric". The most common pediatric brain tumours are the low grade gliomas.

We updated the sentence as suggested.

Line 21: Add "brain" after "embryonal".

We updated the sentence as suggested.

Line 25: Replace "and" with "with" since it is the concurrent "high/high" state that is prognostic.

We updated the sentence as suggested.

Line 56: Replace "were" with "are".

We updated the sentence as suggested.

Lines 70, 73 and 74: Change "investigate" to "investigated".

We updated the sentence as suggested.

Line 96: Of the 9 MB samples, how many were SHH? WNT? Group 3? Group 4?

We added the following sentence in the paragraph: “The 9 MB samples consisted of 5 Group 4 samples, 2 SHH samples, and 2 WNT samples.”

Line 98: Did you mean "papillary"?

We corrected it as “papillary glioneuronal sample”.

Line 193: Add "and" prior to "identified".

We updated the sentence as suggested.

Line 198: Add "of" after "rest".

We updated the sentence as suggested.

Line 199: Add "a" prior to "p-value".

We updated the sentence as suggested.

Line 201: Add "the" after "with".

We updated the sentence as suggested.

Line 232: Replace "is" with "was".

We updated the sentence as suggested.

Line 307: The sentence is unclear as written. What is meant by "not significantly associated with any particular brain tissue"?

We revised the sentence as “the let-7 expression in MB was not significantly different from its expression level in other tissues (MB vs. other tissues, t-test p=0.65, Fig. 4B).”

Line 320: Spell the second use of "let-7" correctly.

We corrected the misspelling.

Line 384: Change to "St. Jude Children's Research Hospital" group.

We updated the sentence as suggested.

Line 386: Replace "is" with "a"

We updated the sentence as suggested.

Line 392: Rewrite as "observations in the GSE85218 data".

We updated the sentence as suggested.

Lines 464-465: This sentence is very confusing as written: "worse survival outcomes identified with worse survival".

We revised the sentence as “SHH-α and SHH-β, subtypes of SHH-MB by SNF, were enriched for MYCN amplifications and were associated with worse survival outcomes”.

Line 499: Revise to "validated".

We updated the sentence as suggested.

Reviewer 3 Report

Overall, this study examined whether let-7 miRNA activity and NMYC expression could predict overall survival in patients with MB and whether tumor cell composition could be used to estimate MB subgroup-specific genomic features. The authors report that increased let-7 activity and high NMYC expression were associated with a worse prognosis. Further, they report that cis-methyl genes could better separate MB subgroups when compared to all genes and that LIN28B expression displayed subgroup variation whereas expression of LIN28A did not. While the authors provide some evidence indicating the prognostic benefit of let-7 activity and NMYC expression to characterize heterogeneous SHH tumors into prognostic subtypes, there are substantial flaws in the report as indicated below.

  1. A central argument of this proposal is that let-7 miRNA activity could potentially be used as a prognosis marker for SHH MB. Although the authors state that there is a strong negative correlation between MYCN expression and let-7 miRNA activity, this correlation does not imply a regulatory role of let-7 miRNA in the MYCN/LIN28B/let-7 axis. Moreover, this axis has been studied primarily in neuroblastoma and ovarian cancer. Whether this axis has any effect in medulloblastoma has yet to be confirmed.
  2.  Importantly, the authors state that let-7 miRNA activity was significantly higher in MB compared to other tissues. However, let-7 miRNA activity was calculated from the mean expression of all let-7 miRNAs. Since the expression of let-7 miRNA was not significantly different when compared to other tissues (Fig. 4B), it cannot be reasonably concluded that let-7 activity is indeed elevated in MB. Surprisingly, the authors even acknowledge this in the legend of figure 4.
  3.  Given that the authors frequently assumed that their calculated correlations by any means suggest a regulatory relationship among let-7 miRNA, MYCN, and LIN28B and that the computations to determine let-7 activity are inherently flawed, there is insufficient evidence to argue for the prognostic value of let-7 miRNA activity for SHH MB. 
  4. The subgroup-specific expression of LIN28B is not convincing. Even if it were, the low LIN28B expression in light of high MYCN expression is insufficient to suggest that let-7 miRNA activity accounts for this discrepancy; several studies have shown that the regulation of let-7, MYCN, and LIN28B involve numerous signaling pathways and effectors.
  5. It is clear that the legends and materials & methods section were not proofread for spelling errors. 
  6. The supplementary figure 2 is unclear. In the methods, the authors simply state that they calculated the correlations between gene expression and activity but fail to provide how these computations were carried out in a clear manner. Further, the caption for supplementary figure 2 is vague; it states that the purity indicates high vs low whereas, in the methods, the labeling of the four groups is different.

Author Response

Overall, this study examined whether let-7 miRNA activity and NMYC expression could predict overall survival in patients with MB and whether tumor cell composition could be used to estimate MB subgroup-specific genomic features. The authors report that increased let-7 activity and high NMYC expression were associated with a worse prognosis. Further, they report that cis-methyl genes could better separate MB subgroups when compared to all genes and that LIN28B expression displayed subgroup variation whereas expression of LIN28A did not. While the authors provide some evidence indicating the prognostic benefit of let-7 activity and NMYC expression to characterize heterogeneous SHH tumors into prognostic subtypes, there are substantial flaws in the report as indicated below.

We respectfully disagree with a few points in the summary.  Figure 1 shows that methylation-driven transcriptomic variations can partition MB samples in to four subgroups, consistent with the knowledge that the four subgroups are derived from different progenitor cells which have distinct methylation profiles.  Based on the methylation-driven transcriptomic variations, we are able to estimate tumor cell proportions in bulk tissues, opposite from the reviewer summary “whether tumor cell composition could be used to estimate MB subgroup-specific genomic features”.

  1. A central argument of this proposal is that let-7 miRNA activity could potentially be used as a prognosis marker for SHH MB. Although the authors state that there is a strong negative correlation between MYCN expression and let-7 miRNA activity, this correlation does not imply a regulatory role of let-7 miRNA in the MYCN/LIN28B/let-7 axis. Moreover, this axis has been studied primarily in neuroblastoma and ovarian cancer. Whether this axis has any effect in medulloblastoma has yet to be confirmed.

We agree with the reviewer that the MYCN/LIN28B/let-7 axis has not been extensively studied in medulloblastoma.  Rodini et al (J. Neuto-Oncology, 2012) shows that high LIN28 expression is significantly correlated with shorter survival in medulloblastoma in general. Hovestadt et al (Nature, 2014) shows that LIN28B regulates multiple oncogenic processes, in part by downregulating LET-7 miRNA family in medulloblastoma and LIN28B expression is associated with survival in MB patients with Group 3 and Group 4 subtypes.  Whether LIN28/let-7 plays an important role in SHH-MB is not known in literature.  However, this is not a flaw of the study.  This actually highlights the novelty of our findings.

We also agree with the reviewer that a negative correlation between MYCN expression and let-7 miRNA activity does not directly imply a causal regulatory relationship between them.  In vitro and/or in vivo perturbation experiments are needed to validate the causal relationship.  As the reviewer 2 pointed out, there are multiple steps needed from discoveries to clinical impact.  Our findings are just the start of the long process.

  1.  Importantly, the authors state that let-7 miRNA activity was significantly higher in MB compared to other tissues. However, let-7 miRNA activity was calculated from the mean expression of all let-7 miRNAs. Since the expression of let-7 miRNA was not significantly different when compared to other tissues (Fig. 4B), it cannot be reasonably concluded that let-7 activity is indeed elevated in MB. Surprisingly, the authors even acknowledge this in the legend of figure 4.

The method of inferring miRNA activity, ActMiR (Lee et al, Bioinformatics, 2016; Lee et al, Oncogene, 2018), is based on expression levels of miRNA predicted target genes in a tissue.  Under an ideal condition, expression level and activity of a miRNA should be tightly correlated.  In reality, the two are different as we showed in the above two papers. Many experimental studies reveal potential molecular mechanisms driving the deviation of miRNA activity and expression level, such as miRNA sponges (review by Ebert and Sharp, Current Biology, 2021; Mullokandov et al, Nature Methods, 2012).

We added the following sentence at the beginning of the Materials and Method section 2.5: “We previously developed a tool for inferring miRNA activity, ActMiR [46,56], based on expression levels of miRNA predicted target genes in a tissue.”

We are sorry to have the confusing sentence in the legend of Figure 4. We removed the sentence from the figure legend.

  1.  Given that the authors frequently assumed that their calculated correlations by any means suggest a regulatory relationship among let-7 miRNA, MYCN, and LIN28B and that the computations to determine let-7 activity are inherently flawed, there is insufficient evidence to argue for the prognostic value of let-7 miRNA activity for SHH MB. 

We respectfully disagree with the reviewer comment.  As we outlined above, the miRNA active inference method, ActMiR, has been extensively validated in many independent datasets and in vitro perturbation experiments (Lee et al, Bioinformatics, 2016; Lee et al, Oncogene, 2018; Degli et al, Sci Rep, 2017; Aushev et al, Clin Cancer Res, 2018).  The molecular mechanisms driving the differences between miRNA activity and expression level are extensively studied experimentally as described above.  Further validation of our computational method for inferring let-7 activity is out of scope of this study.

We do agree with the reviewer that further experimental validations are needed to confirm regulatory relationships among let-7, MYCN, and LIN28 as we describe in our response to the reviewer’s comment 1 and reviewer 2.

  1. The subgroup-specific expression of LIN28B is not convincing. Even if it were, the low LIN28B expression in light of high MYCN expression is insufficient to suggest that let-7 miRNA activity accounts for this discrepancy; several studies have shown that the regulation of let-7, MYCN, and LIN28B involve numerous signaling pathways and effectors.

Figure 5B shows a significant anti-correlation between MYCN expression and let-7 activity (γ=-0.37, p=1.7x10-8).  We agree with the reviewer that further experimental validations are needed to establish a causal relationship from a correlation.

  1. It is clear that the legends and materials & methods section were not proofread for spelling errors. 

We thank the reviewer pointing out spelling errors.  We updated the text following suggestions from reviewer 1 and 2, and further proof-read the legends and Materials and Methods section.

  1. The supplementary figure 2 is unclear. In the methods, the authors simply state that they calculated the correlations between gene expression and activity but fail to provide how these computations were carried out in a clear manner. Further, the caption for supplementary figure 2 is vague; it states that the purity indicates high vs low whereas, in the methods, the labeling of the four groups is different.

The inference of miRNA activity is described in Methods section 2.5 and tumor purity estimation for SHH-MB is described in Methods section 2.4. Labeling for MYCN expression and let-7 activity was also added to Methods section 2.7 to match the rest of the manuscript. The correlations in Supplementary Figure 2 (now Figure S3) are based on Pearson correlation.  We are sorry the legend for Supplementary Figure 2 was not clear.  We updated the legend as the following:

Figure S3. Partitioning SHH-MB samples into MYCN expression low/high or let-7 activity low/high with tumor purity taken into consideration. We performed linear regression between purity (Methods 2.4) and MYCN expression and between purity and let-7 activity. The line of best fit (representing mean expression level or activity at a tumor purity level) was used to define high vs low purity. High purity was greater than the linear regression line of best fit, and low purity was defined as less than the linear regression line of best fit. A) GSE85218 linear regression for MYCN vs purity. B) GSE85218 linear regression for let-7 activity vs purity. C): St Jude/validation linear regression for MYCN vs purity. D) St Jude/validation linear regression for let-7 activity vs purity”

Round 2

Reviewer 3 Report

Comments and Suggestions for Authors

Overall, this study examined whether let-7 miRNA activity and NMYC expression could predict overall survival in patients with MB and whether tumor cell composition could be used to estimate MB subgroup-specific genomic features. The authors report that increased let-7 activity and high NMYC expression were associated with a worse prognosis. Further, they report that cis-methyl genes could better separate MB subgroups when compared to all genes and that LIN28B expression displayed subgroup variation whereas expression of LIN28A did not. While the authors provide some evidence indicating the prognostic benefit of let-7 activity and NMYC expression to characterize heterogeneous SHH tumors into prognostic subtypes, there are substantial flaws in the report as indicated below.

We respectfully disagree with a few points in the summary.  Figure 1 shows that methylation-driven transcriptomic variations can partition MB samples into four subgroups, consistent with the knowledge that the four subgroups are derived from different progenitor cells which have distinct methylation profiles.  Based on the methylation-driven transcriptomic variations, we are able to estimate tumor cell proportions in bulk tissues, opposite from the reviewer summary “whether tumor cell composition could be used to estimate MB subgroup-specific genomic features”.

  1. A central argument of this proposal is that let-7 miRNA activity could potentially be used as a prognosis marker for SHH MB. Although the authors state that there is a strong negative correlation between MYCN expression and let-7 miRNA activity, this correlation does not imply a regulatory role of let-7 miRNA in the MYCN/LIN28B/let-7 axis. Moreover, this axis has been studied primarily in neuroblastoma and ovarian cancer. Whether this axis has any effect in medulloblastoma has yet to be confirmed.

We agree with the reviewer that the MYCN/LIN28B/let-7 axis has not been extensively studied in medulloblastoma.  Rodini et al (J. Neuto-Oncology, 2012) shows that high LIN28 expression is significantly correlated with shorter survival in medulloblastoma in general. Hovestadt et al (Nature, 2014) shows that LIN28B regulates multiple oncogenic processes, in part by downregulating LET-7 miRNA family in medulloblastoma and LIN28B expression is associated with survival in MB patients with Group 3 and Group 4 subtypes.  Whether LIN28/let-7 plays an important role in SHH-MB is not known in literature.  However, this is not a flaw of the study.  This actually highlights the novelty of our findings.

We also agree with the reviewer that a negative correlation between MYCN expression and let-7 miRNA activity does not directly imply a causal regulatory relationship between them.  In vitro and/or in vivo perturbation experiments are needed to validate the causal relationship.  As the reviewer 2 pointed out, there are multiple steps needed from discoveries to clinical impact.  Our findings are just the start of the long process.

In the discussion, the authors convey that future in vitro and in vivo experiments will substantiate their work, addressing the concern regarding the regulatory role of let-7 miRNA in the MYCN/LIN28B/let-7 axis as it relates to MB.

  1.  Importantly, the authors state that let-7 miRNA activity was significantly higher in MB compared to other tissues. However, let-7 miRNA activity was calculated from the mean expression of all let-7 miRNAs. Since the expression of let-7 miRNA was not significantly different when compared to other tissues (Fig. 4B), it cannot be reasonably concluded that let-7 activity is indeed elevated in MB. Surprisingly, the authors even acknowledge this in the legend of figure 4.

The method of inferring miRNA activity, ActMiR (Lee et al, Bioinformatics, 2016; Lee et al, Oncogene, 2018), is based on expression levels of miRNA predicted target genes in a tissue.  Under an ideal condition, expression level and activity of a miRNA should be tightly correlated.  In reality, the two are different as we showed in the above two papers. Many experimental studies reveal potential molecular mechanisms driving the deviation of miRNA activity and expression level, such as miRNA sponges (review by Ebert and Sharp, Current Biology, 2021; Mullokandov et al, Nature Methods, 2012).

We added the following sentence at the beginning of the Materials and Method section 2.5: “We previously developed a tool for inferring miRNA activity, ActMiR [46,56], based on expression levels of miRNA predicted target genes in a tissue.”

We are sorry to have the confusing sentence in the legend of Figure 4. We removed the sentence from the figure legend.

The authors included references to their previous work using ActMiR and made changes made to the legend (Figure 4), both of which address concerns regarding data processing and interpretation.

  1.  Given that the authors frequently assumed that their calculated correlations by any means suggest a regulatory relationship among let-7 miRNA, MYCN, and LIN28B and that the computations to determine let-7 activity are inherently flawed, there is insufficient evidence to argue for the prognostic value of let-7 miRNA activity for SHH MB. 

We respectfully disagree with the reviewer comment.  As we outlined above, the miRNA active inference method, ActMiR, has been extensively validated in many independent datasets and in vitro perturbation experiments (Lee et al, Bioinformatics, 2016; Lee et al, Oncogene, 2018; Degli et al, Sci Rep, 2017; Aushev et al, Clin Cancer Res, 2018).  The molecular mechanisms driving the differences between miRNA activity and expression level are extensively studied experimentally as described above.  Further validation of our computational method for inferring let-7 activity is out of scope of this study.

We do agree with the reviewer that further experimental validations are needed to confirm regulatory relationships among let-7, MYCN, and LIN28 as we describe in our response to the reviewer’s comment 1 and reviewer 2.

  1. The subgroup-specific expression of LIN28B is not convincing. Even if it were, the low LIN28B expression in light of high MYCN expression is insufficient to suggest that let-7 miRNA activity accounts for this discrepancy; several studies have shown that the regulation of let-7, MYCN, and LIN28B involve numerous signaling pathways and effectors.

Figure 5B shows a significant anti-correlation between MYCN expression and let-7 activity (γ=-0.37, p=1.7x10-8).  We agree with the reviewer that further experimental validations are needed to establish a causal relationship from a correlation.

  1. It is clear that the legends and materials & methods section were not proofread for spelling errors. 

We thank the reviewer pointing out spelling errors.  We updated the text following suggestions from reviewer 1 and 2, and further proof-read the legends and Materials and Methods section.

The authors have addressed spelling issues; please check Table 5 as it mentions “HH-MB samples”

  1. The supplementary figure 2 is unclear. In the methods, the authors simply state that they calculated the correlations between gene expression and activity but fail to provide how these computations were carried out in a clear manner. Further, the caption for supplementary figure 2 is vague; it states that the purity indicates high vs low whereas, in the methods, the labeling of the four groups is different.

The inference of miRNA activity is described in Methods section 2.5 and tumor purity estimation for SHH-MB is described in Methods section 2.4. Labeling for MYCN expression and let-7 activity was also added to Methods section 2.7 to match the rest of the manuscript. The correlations in Supplementary Figure 2 (now Figure S3) are based on Pearson correlation.  We are sorry the legend for Supplementary Figure 2 was not clear.  We updated the legend as the following:

Figure S3. Partitioning SHH-MB samples into MYCN expression low/high or let-7 activity low/high with tumor purity taken into consideration. We performed linear regression between purity (Methods 2.4) and MYCN expression and between purity and let-7 activity. The line of best fit (representing mean expression level or activity at a tumor purity level) was used to define high vs low purity. High purity was greater than the linear regression line of best fit, and low purity was defined as less than the linear regression line of best fit. A) GSE85218 linear regression for MYCN vs purity. B) GSE85218 linear regression for let-7 activity vs purity. C): St Jude/validation linear regression for MYCN vs purity. D) St Jude/validation linear regression for let-7 activity vs purity”

The authors have thoroughly clarified the concern of miRNA activity along with tumor purity estimation by making changes to the legend of Supplementary Figure 2 (changed to Figure S3).  

The authors have clarified the concern of miRNA activity along with tumor purity estimation by making changes to the legend of Supplementary Figure 2 (changed to Figure S3). The authors included references to their previous work using ActMiR and made changes made to the legend (Figure 4), both of which address concerns regarding data processing and interpretation. The discussion section was improved as the authors convey that future in vitro and in vivo experiments will substantiate their work, addressing the concern regarding the regulatory role of let-7 miRNA in the MYCN/LIN28B/let-7 axis as it relates to MB. The authors have addressed spelling issues; please check Table 5 as it mentions "HH-MB samples"

Author Response

We thank the reviewer for carefully reading through the manuscript.  We corrected the typo.